# Wearable Biosensor Technology in Education: A Systematic Review

**DOI:** 10.3390/s24082437

**Published:** 2024-04-11

**Authors:** María A. Hernández-Mustieles, Yoshua E. Lima-Carmona, Maxine A. Pacheco-Ramírez, Axel A. Mendoza-Armenta, José Esteban Romero-Gómez, César F. Cruz-Gómez, Diana C. Rodríguez-Alvarado, Alejandro Arceo, Jesús G. Cruz-Garza, Mauricio A. Ramírez-Moreno, Jorge de J. Lozoya-Santos

**Affiliations:** 1Mechatronics Department, School of Engineering and Sciences, Monterrey Campus, Tecnologico de Monterrey, Monterrey 64700, Mexico; maria.hernandezm@tec.mx (M.A.H.-M.); a01734063@exatec.tec.mx (Y.E.L.-C.); mpachec4@cougarnet.uh.edu (M.A.P.-R.); axel.mendoza@tec.mx (A.A.M.-A.); a00827233@tec.mx (C.F.C.-G.); a00832786@tec.mx (D.C.R.-A.); alejoarceo@tec.mx (A.A.); mauricio.ramirezm@tec.mx (M.A.R.-M.); 2Mechatronics Department, School of Engineering and Sciences, Guadalajara Campus, Tecnologico de Monterrey, Guadalajara 45201, Mexico; a01639031@tec.mx; 3Department of Neurosurgery, Houston Methodist Research Institute, Houston, TX 77030, USA; jgc243@cornell.edu

**Keywords:** biometrics, education, neuroeducation, wearable biosensor technology

## Abstract

Wearable Biosensor Technology (WBT) has emerged as a transformative tool in the educational system over the past decade. This systematic review encompasses a comprehensive analysis of WBT utilization in educational settings over a 10-year span (2012–2022), highlighting the evolution of this field to address challenges in education by integrating technology to solve specific educational challenges, such as enhancing student engagement, monitoring stress and cognitive load, improving learning experiences, and providing real-time feedback for both students and educators. By exploring these aspects, this review sheds light on the potential implications of WBT on the future of learning. A rigorous and systematic search of major academic databases, including Google Scholar and Scopus, was conducted in accordance with the PRISMA guidelines. Relevant studies were selected based on predefined inclusion and exclusion criteria. The articles selected were assessed for methodological quality and bias using established tools. The process of data extraction and synthesis followed a structured framework. Key findings include the shift from theoretical exploration to practical implementation, with EEG being the predominant measurement, aiming to explore mental states, physiological constructs, and teaching effectiveness. Wearable biosensors are significantly impacting the educational field, serving as an important resource for educators and a tool for students. Their application has the potential to transform and optimize academic practices through sensors that capture biometric data, enabling the implementation of metrics and models to understand the development and performance of students and professors in an academic environment, as well as to gain insights into the learning process.

## 1. Introduction

There has been a significant surge and evolution in research on Wearable Biosensor Technology (WBT) in recent years [1], along with its integration into educational environments. WBT refers to a subset of wearable technology devices that are designed to be worn directly or loosely by an individual and that are equipped with an arrangement of built-in sensors that allow for the acquisition of physiological or biometric data [2]. The wide applicability of these technologies ranges from healthcare (for treatment, rehabilitation, or monitoring) [3] and safety (for fall detection and fall prevention, fatigue detection and environmental condition monitoring) [4] to activity recognition in sports [5] and education [6], among others.

Nowadays, WBT has been used in educational contexts to enhance the learning experience and study the effects of its incorporation [7,8]. WBTs have been used to guide the structure of learning programs, capture data to inform the process of learning, make knowledge visible, and help instructors learn about their students [9]. One of the first documented cases of the use of wearable devices in education incorporated the use of virtual reality (VR) technology for mathematics and geometry education with the help of a tutor in the virtual space [10]. In recent years, smartwatch devices have been the focus of interest due to their unique features, such as their comfortable portability and the ability to support learning and everyday activities [11,12,13]. Currently, smartwatches have been recognized as promising in educational contexts given their growing acceptance and adoption as a personal wearable device [14]. Other applications of WBT include identity management systems, class attendance, e-evaluation, security, student motivations, and learning analytics [15]. The biometric technology market is expected to reach a value of USD 94 billion by 2025 at a compound annual growth rate of 36% [15], when just 10 years prior, in 2015, it was valued at USD 9.916 million [16]. This increase in market value points to a growth in the development and acceptance of this type of technology.

The adoption of WBTs in education provides several advantages. One of the main benefits of adopting WBT is its ability to facilitate convenient access and interaction with biometric information and learning materials with little restrictions regarding time and place of access [17]. Students and teachers can benefit from this information by accessing learning materials at any time and any place while also guaranteeing valuable data collection in various educational settings for subsequent analysis [11]. This would reflect a non-restrictive, unobtrusive learning experience for students. A clear example of this can be found in [18,19], where WBTs are incorporated into tasks for physical activity recognition and biomechanical feedback applications, respectively, to improve students’ sports performance and health.

When combined with other tools such as the Internet of Things (IoT), smartwatches, and eye-tracking technology, wearables can be used to estimate student attention [20]. WBT can also be used to implement performance evaluation systems [21] or emotion recognition systems for students with different needs, for instance, those who present a mental disorder or mood disruption [22].

A second benefit of adopting WBT in education is the value of the implicit information offered by the collected physiological data. In [23], the term “neurophysiological measurement” is introduced, which refers to an exclusive type of physiological data that are related to the Central Nervous System (CNS) or the Autonomic Nervous System (ANS). On this note, neurophysiological measurements (NPMs) related to the ANS include measurements such as eye-related measurements (blink rate and pupil dilation), electrodermal activity (EDA) or galvanic skin response (GSR), blood pressure, and electrocardiography (ECG), while NPMs related to the CNS include electroencephalography (EEG) and electromyography (EMG) [23]. From this list, EEG is of particular interest in an educational context as it measures brain activity, which can be used to infer fluctuations in cognitive processes [24,25]. It is widely known that psychological constructs such as cognitive load, attention, and emotion play an important role in the learning process of a student [23]. NPMs such as EEG, heart rate variability (HRV), or EDA can provide valuable neurological data to monitor mental states and determine a student’s performance [26,27,28,29,30].

Figure 1 shows a summary of the physiological measurements considered for this review, along with some of the devices used to acquire them. The combination of such measurements with machine learning (ML) algorithms can aid in the detection of low academic performance and is useful for deciding preventive actions [31,32]. Additionally, integrating VR technology has allowed for the design and testing of different learning environments with more convenience and the study of how they affect cognitive processes in students [33,34,35].

A third benefit of the use of WBT in education is that the monitoring of NPMs can be exploited to solve educational challenges. They can be used to predict cognitive outcomes such as students’ academic performance by using peer-to-peer or student–teacher brain-to-brain (B2B) synchronization and interaction [36,37,38,39]. This allows for an increase in the effectiveness of teaching and learning processes [23].

The purpose of this review is to critically examine the existing literature to assess the impact of the application of WBT in education and the limits that it encompasses. We aim to investigate the evolution of WBT in education over the past 10 years, how it has been integrated to solve key educational challenges, the wide range of educational areas in which it can be applied, and the future perspectives, challenges, and trends for this technology. A detailed discussion over the evolution, trends, applications, and challenges of WBT in education is presented in order to provide a guide for future research in this field.

The rest of this article is divided as follows: Section 2 describes the methodology used to write this review; Section 3 presents the evolution of WBT in education, state-of-the-art implementations, and current applications in the field; Section 4 discusses the challenges and current trends in this technology and provides perspectives; and finally, Section 5 closes the article with the conclusions of this work.

## 2. Materials and Methods

### 2.1. Study Design and Search Strategy

A systematic search, following the (Preferred Reporting Items for Systematic Reviews and Meta-Analyses) PRISMA methodology [40] was applied in this review. The literature review took place on 21 September 2022 within the Scopus database and Google Scholar; Scopus is widely acknowledged as one of the main bibliographic databases, distinguished by its extensive content coverage and robust impact indicators [41]. This selection affords access to valuable metrics crucial for data analysis and the construction of the content distribution presented, allowing us to complement it correctly with the literature consulted on Google Scholar, considering publications that were available within the January 2012 to August 2022 period. The following string represents the equation formulated by the relevant keywords related to all aspects of WBT in education:


*(“Biometry” OR “Biometrics” OR “EEG” OR “Electroencephalography” OR “Electroencephalogram” OR “Biofeedback” OR “ECG” OR “Electrocardiogram” OR “BPM” OR “Beats per Minute” OR “Blood Volume Pulse” OR “HRV” OR “Heart Rate Variability” OR “Devices” OR “Sensors” OR “Smartwatch” OR “Wearable”) AND (“Education” OR “Remote Education” OR “Learning” OR “e-learning” OR “Student” OR “Teacher” OR “Professor” OR “Teaching” OR “Classroom” OR “School Activity” OR “Academic Task” OR “Exam” OR “Academic” OR “Learning Outcomes" OR “Reading Comprehension”) AND (“Mental Fatigue” OR “Stress” OR “Cognitive Workload” OR “Applications” OR “Perspectives” OR “Limitations” OR “Challenges” OR “Innovation” OR “Advantages” OR “Disadvantages” OR “Technology”) AND NOT (“Deep Learning”) AND NOT (“Machine Learning”) AND NOT (“Reinforcement Learning”).*


### 2.2. Exclusion Criteria

Studies were excluded if they met one or more of the criteria in the following list:The publication was not related to biometry or education (n = 96).The publication was related to biometry but not to education (n = 57).The publication was related to education, but not to biometry (n = 145).The search was related to a summary of conference proceedings (n = 3).

## 3. Results

### 3.1. Summary of Studies Included

A total of 368 works were detected in Scopus using the equation presented in Section 2. Duplicates reported from the database and studies within the exclusion criteria were discarded. From the identified papers, 301 studies were eliminated due to falling within the exclusion criteria, and only 66 were considered. Additionally, 74 studies were also included from citation searching in Google Scholar. A summary of the results obtained from the search is shown in Figure 2.

### 3.2. General Characteristics of the Included Studies

The general characteristics of the 66 included studies from Scopus are summarized in Table 1. This table presents the following characteristics for each study:**Objective.** Describes the main goal of the study being conducted.**Education Type.** Classifies the study according to the type of education to which it is applied, such as academic, language, medical, Science Technology Engineering Mathematics (STEM), etc.**Education Level.** Classifies the study according to the level of education to which it is applied, such as kindergarten, elementary school, high school, university, etc.**Institute.** Provides the name of the institution in which the study is being conducted.**Country.** Provides the name of the country in which the study was conducted.**Sample Size.** Number of people who participated as test subjects during the study.**Analysis Tools.** Provides information on the tools used to gather and analyze the study’s data. The information collected in each study includes mainly physiological characteristics, such as EEG, ECG, EMG, HR, GSR, and HRV, and some questionnaires such as the Medical Student Stressor Questionnaire (MSSQ), Perceived Stress Scale (PSS-10), Behavior Assessment System for Children (BASC-S2), Global Assessment of Recent Stress (GARS-K), Balance of Challenge and Skill (BCS), and Momentary Test Performance (MOM-tp). On the other hand, a diverse set of tools was used to analyze the information, including MATLAB, Statistical Package for the Social Sciences (SPSS), augmented reality (AR), VR, wearable commercial-off-the-shelf (COTS), and brain–computer interfaces (BCIs). Lastly, in order to provide reliable results, the studies employed various types of metrics or statistics, which included standard deviation of NN intervals (SDNN), root mean square of successive differences between normal heartbeats (RMSSD), proportion of NN50 (pNN50), low frequency (LF) and high frequency (HF) ratio, ANOVA, radial basis neural network (RBFNN), and improved extreme learning machine (IELM).**Contribution.** Contains the main findings of the study.

It is important to note that all of the devices presented in this review only take recordings during experimental settings and for a limited period of time. Studies about continuous monitoring devices are not included.

### 3.3. Temporal Distribution of the Included Studies

Figure 3 shows the temporal distribution of the selected papers from Scopus and Google Scholar that were published from 2012 to 2022. During this period, an increasing trend in the implementation and exploration of WBTs in the field of education can be observed. This illustrates how researchers, scientists, and scholars have adapted to new challenges, harnessed emerging technologies, and forged pathways to address the complexities of the educational system during the last decade.

Furthermore, this increasing trend observed from 2012 to 2022 can be attributed to different factors, since, during the first years of research (2012 to 2017, mean: 4.83 studies, std: 3.92), the theoretical part and the practical basis of the field were established. Following this, starting in 2018 until 2022 (mean: 22.20 studies, std: 4.60), there was a growing recognition of the importance of these technologies in education as they became more sophisticated and more accessible.

In summary, the temporal graph of publications related to the implementation of WBT in education shows a four-fold increase in the number of published articles per year from 2018 to 2022 compared to the articles published between 2012 and 2017. This reflects an increased focus on the convergence of technology and education, which promises significant advances in improving the quality of teaching and learning over the next decade.

### 3.4. Geographical Distribution of the Included Studies

Figure 4 shows the geographical distribution of the included studies from 2012 to 2022. It reveals a diverse and widespread interest in WBT in education across the globe. Notably, China emerges as a pioneer in this field, with 20 studies contributing valuable insights. Following closely, the United States demonstrates significant engagement with 18 studies, underlining its prominent role in advancing research in this area. Mexico also surfaces as a noteworthy participant, with 10 studies, highlighting a growing interest in wearable biosensors within the educational context.

The collective picture is truly international, with a total of 45 countries actively contributing to the body of knowledge on WBTs in education during the specified time period. This extensive global involvement underscores the universal significance and appeal of WBT in shaping educational practices. As diverse nations collaborate and contribute, it fosters a rich and comprehensive understanding of the implications and applications of this technology in enhancing educational methodologies worldwide.

### 3.5. Literature Review

#### 3.5.1. Evolution of WBT in Education

It has been observed over the years that, for educational institutions, it is difficult to extract information that helps understand the way students learn, as well as to guarantee enhancing learning experiences [86], taking into account the challenges represented by teaching people with different educational backgrounds and learning engagement styles [87]. Educational institutions have been incorporating and implementing new gadgets like wearable and mobile devices, making it easier to obtain data from students in order to improve how they learn by making data-based changes to their infrastructure or teaching methodologies [88,89].

In July 2012, a study was conducted in which EEG was used to estimate and predict mathematical problem-solving outcomes. This study aimed to evaluate whether estimates of the attention and cognitive workload of students obtained from recorded EEG data while they solved math problems could be useful in predicting success or failure. The signals were processed to obtain the mental states of students in the frequency domain. Based on the results obtained from a Support Vector Machine (SVM) model, the transitions between different state levels can predict problem-solving outcomes with an average accuracy of 62 percent for both easy and hard difficulties [90].

As another example of these types of implementations, in 2018, Hui Zheng and Vivian Genaro Motti [91] created “WELI” to investigate how smartwatches can support students with Intellectual and Developmental Disabilities (IDDs). The goal was to help students with IDDs in the performance of activities requiring high emotional and behavioral skills, as well as involvement, communication, collaboration, and planning. Furthermore, in 2017, a multibiometric system was developed, aimed at authenticating students on online learning platforms. This algorithm verifies the presence and interaction of students by calculating the score-level fusion of different biometric responses. This system serves as a tool to accredit the identity of the person undergoing the learning experience [92].

In 2019, a paper showed an implementation of adaptability and artificial intelligence (AI) methods within the Education 4.0 framework and also investigated the embedded biosensors used in smartphones and smartwatches [93]. In this context, Education 4.0 is the integration of emerging technologies such as analytics, AI, biometrics, and the IoT within the educational framework in preparation for the industry. They proposed a framework for education that uses embedded biosensor data (EMG, EDA, ECG, blood pressure, and EEG) and environmental data to estimate students’ well-being and health. Recent studies have continued to explore learning/Education 4.0 by exploring emotional and cognitive engagement classification through EEG [94]. This study classified states of low/high engagement with a 77% accuracy.

In another study, the authors developed a BCI for gathering data and detecting a learner’s mental state while watching MOOC (Massive Open Online Course) videos through EEG devices. Their proposal was based on John Sweller’s cognitive load theory to develop a model with preprocessed training data and test the classifiers to validate their ensemble classifiers’ performance [95]. Other studies have continued to explore the approach of assessing a learner’s engagement and attention during video lectures through inter-subject metrics [96].

During the recent COVID-19 pandemic, the University of Pamplona, in Colombia, conducted a research study where they measured EDA, ECG, and EMG in an academic context during stressful situations. This was a study for the detection and identification of the volatile organic compound profiles emitted by the skin. The aim was to measure the student’s stress state during the exam and during the relaxation state, after the exam period [55].

New developments have not only occurred in hardware, but new software and processing techniques have also emerged. In 2022 [48], a study found better classification results from EEG data as a predictor of student stress through the use of an improved extreme learning machine model. A useful approach for EEG processing uses traditional SVMs whose features were extracted through empirical mode decomposition to obtain a higher classification accuracy in predicting student interest [97]. Another metric that has already been used in real-world applications, but is still being developed, is the B2B synchrony measured through EEG [36,98]. Software advancements have also been implemented to enable adaptive learning to, for example, provide video feedback to increase engagement upon the detection of low attention via EEG [99].

As it is evidenced in Figure 5, a wide variety of biosensors have been used in education with diverse applications [100]. Another study [101] identified EEG, ECG, EMG, skin temperature (ST), photoplethysmography (PPG), GSR, and EDA as some of the main physiological signals obtained by sensors to monitor students’ engagement. In the early 2000s, a trend regarding the use of e-textiles in educational contexts appeared, but almost all data were related to posture, gestures, and respiratory patterns. Wearables for learning purposes reached peak development around 2014–2016, when technological advances, such as smart wristbands, watches, and glasses, arrived with the possibility of acquiring precise physiological data [9]. In recent years, there has been a notable surge in technological progress, marked by the emergence of solutions employing more advanced algorithms and machine learning techniques [34,35,101]. These innovations are designed to efficiently process vast amounts of data, addressing specific problems within defined scenarios.

#### 3.5.2. Solving Educational Problems with WBTs

The educational system has been integrating new methods and techniques to improve how students learn. Educational demands change over time, and institutions have to adapt their teaching methods to ensure an optimal learning process. Technology development increases continuously and, as a result, new technologies have allowed for the monitoring of students while they learn that give feedback on the efficiency of teaching methodologies [102,103].

**Figure 5 sensors-24-02437-f005:**
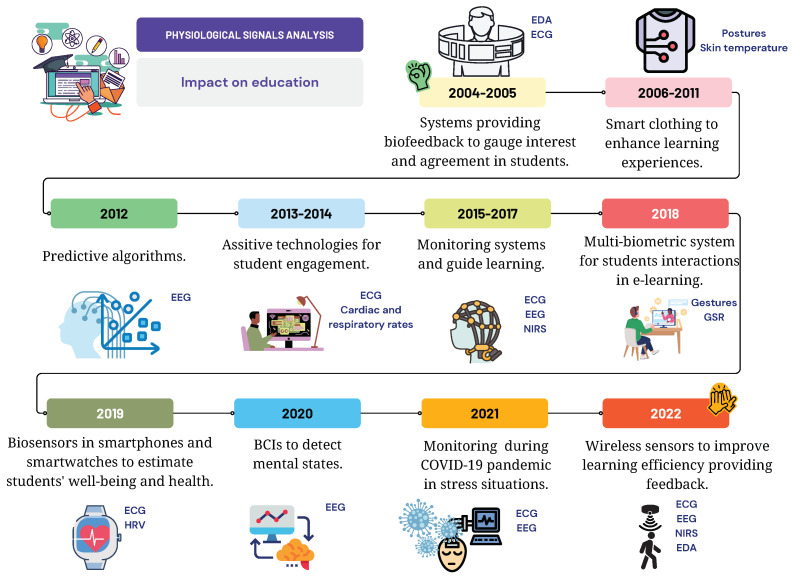
Significant progress timeline of WBT evolution in educational contexts from 2004 to 2022.

WBT has been useful in the academic field in various aspects [15], considering that emotional states and cognitive status are considered good metrics to be aware of the student’s academic progress [104]. Having access to this kind of data allows teachers to identify motivations and optimize the learning process. In this respect, HRV monitoring shows a good performance regulating emotional state, as six breaths per minute are shown to reduce stressful emotions and contribute to improved learning experiences [43], but techniques to characterize cognitive statuses are still being studied. Additionally, WBTs can save institutional resources, optimizing systems like access points, transportation, and other control criteria, which not only has an impact on education but also on safety and security [105].

A high academic load often drives students to develop coping behaviors. EEG recordings during exam situations can serve as adequate indicators of adaptive responses as frontal cortex activation correlates with brain processes that support motivational systems. Stressful situations, such as coping behavior, may push students towards less effective ways of handling the situation [106]. In this context, neurofeedback represents a growing opportunity to monitor mental states. For this reason, various universities tested adaptive neuro-learning systems using a BCI for online education, showing an enhanced learning performance (average test scores of 83.83 out of 100 for the experimental group compared to 56.67 for the control group) [99]. Considering that changes in EEG alpha asymmetry have been observed in the prefrontal cortex depending on the approach or avoidance of motivational systems using positive or negative effects in students, this has demonstrated how positive traits lead to left hemispheric activation, influencing the adaptive response of brain processes and manifesting in an improved academic performance [106].

Specific studies have been developed to solve different problems in education regarding intellectual disabilities. The implementation of a monitoring system using EEG, ECG, and near-infrared spectroscopy (NIRS) offers a valuable tool for assessing cognitive states, in this case, to measure the educational effect on children with mental retardation over four years [107]. In 2022 [43], a study to reduce anxiety and social stress in primary students was released. It shows how having instant biofeedback of the heart rate variability allows for the teaching of an easier method of conscious breathing, leading in consequence to a positive impact on the emotional experience of the students who know how to perform slow and steady breathing.

Given that cognitive load is a fundamental factor in cognitive processing and has a significant impact on clinical reasoning, a study that recorded ECG signals from students at the Uniformed Services University of the Health Sciences was able to identify a correlation between cardiovascular measures and activities associated with high levels of cognitive load [29]. This leads to the conclusion that this type of feedback can aid in enhancing instructional materials and, in turn, improve the future performance of medical students while reducing cognitive load. Using similar physiological measurement techniques with ECG, a study was conducted on college students. In this case, the objective was to analyze how the environment affects students’ learning performance and their psychophysiological responses depending on thermal conditions. The results showed that ECG measurements served as objective indicators to control the task’s load [30].

Understanding the relevance of the fields of STEM in industry settings and assessing vocational interests in these areas can be a complex task, traditionally achieved through various psychometric tests. However, it is possible to evaluate these interests using EEG data [108]. A study was conducted to evaluate the performance of children in topics offered by machine care education (children’s education in STEM), such as programming, 3D design, and robotics. This study aimed to demonstrate how the development of a machine learning algorithm, capable of analyzing physiological signals (HRV, EDA, and EEG), can predict an individual’s affinity for engineering. Additionally, WBT can promote STEM education and involvement of students by exposing them to fun and engaging hands-on activities related to do-it-yourself electronics for wearable computing [6].

NPMs including brain activity, cardiac function, and skin conductance have been analyzed in various contexts, leading to the development of models capable of classifying mental fatigue. This demonstrates how the use of wearable devices that measure physiological signals can enhance the experiences of students and workers [109]. Depending on the tasks being undertaken, specific autonomic responses are generated by the human body, with adequate machine learning classification extracting ECG and EDA measurements in a non-invasive manner, and it is possible to identify the type of task being performed [110]. EEG and cardiac activity have also been used to address the issue of the effects of different learning and teaching methods on the learning process and cognitive state of students with the hopes of implementing personalized learning experiences in the future [111,112].

Overall, wearable biosensors have served as a guiding structure for learning. All kinds of physiological feedback and data interpretation provide the possibility to construct a framework for students and evaluate user performance, but they are also helpful in supporting current teaching methodologies and how tasks can be managed [9,101]. Biometric systems are still evolving and offer a wide range of applications not just in education, leading to meaningful strategies to enhance human performance [105].

#### 3.5.3. Applications of WBTs in Education

With the ongoing evolution of WBTs, their integration has brought about a profound transformation in the pedagogical landscape, reshaping the methodologies of teaching and learning. WBTs have arisen as powerful tools, offering a wide range of applications that harness NPMs to deepen our understanding of the intricate processes involved in human learning, a trend that can be seen in Figure 6.

Figure 6 shows a graphical description of the contrast between the periods from 2012 to 2016 and from 2018 to 2022, since in more recent years, there has been an increasing trend in the application of wearables in education based on physiological signals. In the case of applications with EEG signals, it is shown that in the period from 2018 to 2022, there was an increase of 133% in studies compared to studies in the period from 2012 to 2018. Furthermore, the application that had the greatest increase, taking into account its relevance in both periods, was the heart monitoring application, which is mainly due to the fact that it benefited from the easy access of society to wearable devices such as smartwatches. Finally, applications related to physiological signals such as EMG or EDA also exhibited an important growth; nevertheless, compared to other physiological signals, they have not been of great interest to researchers.

It is necessary to consider that each of our physiological signals may shed light on distinct facets of the learning process [101]. Many of the applications provide multiple perspectives of how the process of knowledge acquisition occurs in individuals. Recent trends in research suggest that wearables are starting to be implemented within real-time frameworks to provide direct feedback for educators. Current wearables are being used in real time to monitor stress during exams [72,81], predict mental fatigue [102] and concentration levels [20], and identify flow states [26]. In bringing wearables to naturalistic settings, studies may leverage consumer-grade wearables, real-time data processing techniques, or web applications with dashboards for monitoring. This research is often conducted within the umbrella of IoT applications [62,71,93,102].

Below, a summary of the main applications of WBTs within the realm of education (with a specific focus on NPMs) is presented.

**Electroencephalography.** Since learning is a cognitive process that involves changes in brain activity [60], and considering that some methods to measure the levels of attention and engagement in students may be intrusive [96], EEG signals have been of great relevance to researchers in the development of tools, technologies, and methodologies for the benefit of education. One of the first studies to test students in a naturalistic high school setting analyzed attention, self-reported enjoyment, personality traits, and other social and engagement metrics derived from surveys and EEG to discover the relationship with a student’s brain synchrony. This study found statistically significant associations suggesting brain-to-brain synchrony as a useful marker for predicting classroom interactions and engagement [37]. With the use of portable and low-cost EEG devices, the authors were able to take measurements from students throughout many sessions of their semester in a non-laboratory setting. Follow-up publications expanded on this idea to understand how the student–teacher relationship and retention of class content are correlated with closeness and brain-to-brain synchrony [37].

EEG-based technologies can also be used as predictors of cognitive performance [28] by using the alpha/theta ratio and delta band power (which are indicators of mental fatigue and drowsiness). Alongside facial expressions, EEG can be predictive of states of engagement, attention [113], planning [114], shifting [115], and even student effort [116]. Regarding attention, considering that it is the most important factor in learning, protocols have been proposed to classify the levels of attention in educational environments [117]. Other studies have generated offline algorithms to evaluate primary and middle-school children’s STEM interests [118]. Wearable technologies can also make EEG research more approachable and accessible. A study created a research-based laboratory curriculum for undergraduate students to learn about the theoretical foundations of EEG and the different protocols used in research [119].

Another possible application enabled by detecting cognitive states can be biofeedback systems [102]. This study built a system where learners engaged in a task while their biometrics were displayed in a separate interface to the teacher. Afterward, the data were fed to a random forest classification algorithm that could accurately discern states of mental fatigue. Furthermore, NPMs are a useful tool to detect stress and anxiety in students. This could allow for more particular interventions in high-stress situations, such as college evaluations [60,120]. In [121], a review can be found where the effects of stress on education have been studied using EEG signals.

**Electrocardiography, Photoplethysmography, and Heart Rate.** HRV is a commonly used metric to detect stress. HRV is not a single metric, but usually an analysis performed in both the time and frequency domains during varying lengths of time over a heartbeat signal. A study conducted on medical students [53] related HRV to both stress and academic achievement, which showed a positive correlation between these variables. A study [29] attempted to measure how the heart rate and HRV, measured by ECG, related to cognitive load and performance in medical students watching videos of physician–patient interactions and filling out a post-encounter form. This study found positive correlations between cognitive load, HR, and HRV, while performance was negatively correlated with cognitive load measures. A larger study performed during university final evaluations [72] used HRV and HR to measure the changes in stress amongst students of different academic years throughout the exam. In this case, HRV was lowest when stress was released after the exam. It also showed a lack of adaptation techniques amongst undergraduates of different semesters, with only a measurable difference in heart rates present between first-year graduate and undergraduate students. This study required the use of a small ECG device (made by CardioDiagnostic) and electrodes to be placed on the participant’s chest and abdomen during the evaluation.

With a focus on biofeedback and interventions, another study used HRV in elementary-school students to reduce anxiety and social stress [43]. This study used HeartMath EmWave 2021 Pro. Version software and hardware, both of which are consumer-grade non-invasive devices for HRV measurements and stress management. The heart rate by itself has also been used as a physiological measure to improve engagement and motivation of university students by combining wearable data (Fitbit, Apple Watch, or JINS MEME) with data of academic performance [122].

**Electromyography, Electrodermal Activity, and Others.** Combined with HRV, EDA can be used to identify different cognitive tasks that a person is performing [110], which has the potential to improve coordination and performance in a classroom. GSR—a term used interchangeably with EDA that also measures skin conductance—has also been used in studies [34,55,73] to measure academic stress. EMG is highly accurate at detecting stress using measurements from the left and right trapezius muscles and the left and right erector spinae muscles, which all showed higher activity during stress-inducing tasks. This study also used ECG to derive HRV and improve the accuracy of the SVM classifier [123]. Considering that different biometric signals or data are implemented in the academic environment, some studies have opted to use a combination of these to make the learning environment intelligent; such data include heart rate, emotions, and sweat levels [86]. Another study [55] also used the EMG of the upper trapezius muscle, alongside ECG and GSR to differentiate students in a state of stress (during an exam) and relaxation (after the exam). With simple classification methods, such as SVMs and linear discriminant analysis (LDA), this study achieved a high accuracy with these variables, particularly GSR, in classifying stress and relaxation states.

#### 3.5.4. Sensors Used by WBTs in Education

WBTs have evolved to include multiple sensors, which are based on the type of physiological signal to be focused on. Some of the most common sensors that can be found in WBTs are detailed in the following.

**Electroencephalography.** In the case of EEG signals, electrode variations, such as dry or wet electrodes, are mainly implemented.

**Electrocardiography, Photoplethysmography, and Heart Rate.** Infrared PPG ear sensors, noninvasive auditory sensors, heart rhythm scanner PE, AD8232 ECG chip, Ambu WhiteSensor WS, blood pressure sensors, ECG sensors, and oxygen sensors are mainly used.

**Electromyography, Electrodermal Activity, and Others.** IMU sensors, EMG sensors, GSR sensors, acceleration sensors, MOX gas sensors, movement trackers, eye tracking sensors, and light and temperature sensors are mainly used.

Table 2 includes many of these sensors that have been reported in the literature. This table also aims to provide a brief summary of the technologies used in multiple studies. It includes the technical details of the device, such as the communication protocol, type of storage, and whether it used a simulated or experimental signal. Moreover, Table 3 also provides general information about some of the different biometry devices used in education, such as the type of signal they measure, the sensors implemented, their type of data storage, and power supply.

## 4. Discussion

The search results from the present review show that EEG was the most popular NPM among the studies. It was found to be used as a stand-alone measurement or along with other biometrics such as EDA [68], eye tracking [74], ECG [85], or even EMG and blood pressure [84]. Two main objectives were identified regarding the use of EEG in classrooms: to analyze the mental state of a student through the estimation of physiological constructs or to evaluate teaching and learning effectiveness with the help of qualitative or biofeedback strategies [71,74,84,85].

First, physiological responses to stress have been used to evaluate the performance of students in an academic setting. Stress analysis was of particular interest for researchers, especially during exams or tests, to examine the change in studying and learning patterns of students [65]. Overall, it was found that investigation of stress levels improves the quality of academic classes [45,121]. Students’ stress levels increase before examinations and during timed exams [60,69,124], and high levels of stress are correlated with poorer evaluation performance and psychological health problems [75,81]. Other analyzed physiological constructs include motivation [85], flow state [26], concentration [66], and sustained attention [113], where an increase in all of them correlates to improved educational interventions and allows for the possibility of implementation of e-learning platforms through BCIs or AR systems [26,66]. Meanwhile, an increase in mental fatigue was discovered to increase on 8 h school days (or longer), and it was identified as a factor of high concern in high-school education [64]. Some studies also developed algorithms for emotion recognition in teachers [46] and to evaluate psychological stress in students [48,68].

Secondly, to evaluate teaching and learning effectiveness, researchers tested the acceptability of wearable and mobile devices by also implementing qualitative surveys [49,63], and biofeedback strategies were used to evaluate the effectiveness of lessons and judge cognitive errors in students [74,79,84].

HR is shown to be the second most preferred NPM in classrooms. HRV is estimated either through ECG or PPG. Contrary to EEG, which is sometimes used as a stand-alone physiological measurement, these measurements are usually always used in parallel with others, such as motion [52], blood pressure [5,77], eye tracking [20], EEG [85], GSR, EMG, temperature, and respiration [55,56,78,84]. Once more, stress is the main focus of the studies, with the proposal of stress detection and monitoring frameworks based on gender-centered HRV, GSR, and EMG [42,47,55,56,123] evaluations [53,76], and the proposal of stress-reduction techniques [77,78]. Some studies also researched the relationship between stress levels and sleep, where high stress levels proved to be associated with poor sleep behaviors in students [44,88]. It was once again proved that WBT offers pedagogical opportunities [5,72,80,82,84,85] and supports learning activities through the integration of AR, AI, and IoT devices [20,52,62]. Finally, other NPMs found in the studies are temperature [54], motion [58,59,67], EDA [70], GSR [73], electrooculography (EOG), EMG [57], and voice [61].

Figure 7 presents a summary of the results. China and the United States were the top two countries with the most papers published related to wearable technology in education. MATLAB (The Mathworks Inc., Natick, MA, USA) and Python (Python Software Foundation, Beaverton, OR, USA) proved to be the most popular software to perform signal processing, and EEG and ECG were the most popular measurements.

### 4.1. Perspectives

One of the main limitations identified in the studies is the variability of the WBT used. This technology field is characterized by its diversity, with various devices offering different features and capabilities, but this represents a drawback. Comparing results between studies may be challenging, given that researchers may not consistently evaluate the same types of devices. This also opens the possibility of variability in protocols for the usage of this technology, limiting the consistency of results across studies. For example, the NeuroSky MindWave Headset (NeuroSky, San Jose, CA, USA) is shown to be the most used device for EEG recording (Table 2). However, the data-processing techniques vary, as well as the software used for the task [26,63,66,71,74].

Additionally, few studies seem to consider the acceptance and user experience of WBT by students and teachers as an important research variable. Most studies did apply surveys to qualitatively measure stress or attention levels; however, only a few implemented surveys to determine the acceptability of WBT in classrooms [61,63] or others did not implement any type of qualitative measurement at all. From the application of technology readiness models (TRMs) to measure physical education teachers’ perspectives on WBT, it is possible to identify conditions in infrastructure that better accommodate the use of technological innovations that improve physical education and performance [125]. Another study shows that teachers report benefits in the incorporation of WBT in teaching by receiving real-time feedback on students’ cognitive states and representing tools for the implementation of more dynamic studying sessions; however, students stated that they experience several challenges related to the affordability, technical infrastructure, distractibility, security, ethics, and privacy of these technologies [126]. Providing insights into the perspectives of the main stakeholders of these technologies would allow for their seamless adoption and implementation and would offer better performance results.

It is suggested that future research should focus further on enriching WBT application and implementation scenarios, instead of being limited only to the theoretical analysis or evaluation of frameworks. This would increase the robustness of the analysis of the true impact of this technology in teaching, learning, or in any educational context [127]. Finally, collaboration would also play an important role in standardizing data and processing methodologies, facilitating the reproduction of studies and the comparison of their performance in the future. Research community efforts such as the EEG extension of the Brain Imaging Data Structure (BIDS) [128] and the Standard Roadmap for Neurotechnologies [129] provide a standard for the storage and organization of EEG data and the requirements for the standardization of neurotechnologies, respectively, and could be valuable tools in building future efforts to contribute to this technology’s standardization.

### 4.2. Challenges and Trends

WBT is emerging as a game-changing trend that is set to shape the future of learning methods. By harnessing these technologies in educational settings, it is possible to unlock endless possibilities for personalized and immersive learning experiences [28,34,35,102,130]. The exponential advances in this field have developed new ways to improve education, but with this growth comes several challenges that must be addressed to ensure improved learning outcomes [115].

One of the major challenges of this technology field is the extraction of useful and actionable health information from the large volumes of data generated by wearable biosensors [131,132]. Analyzing and interpreting these data require complex algorithms and machine learning techniques to gain meaningful insights [133].

Another obstacle is the consistency and accuracy of NPMs, which are highly dependent on the interface between the biosensing electrode and the human body [134]. Ensuring the accuracy and reliability of the data collected by WBTs is crucial for their effective implementation in educational settings [135].

Furthermore, integrating WBT into the existing educational infrastructure represents a multi-level challenge. It involves not only incorporating big data analysis methodologies and building environments that take advantage of WBT and adapt to the education type presented [136,137,138,139], but also addressing issues related to privacy and data security, as WBTs collect sensitive personal information [80,140,141]. Privacy and security issues are challenges that need to be considered; all biometric information must be obtained with the user’s consent and therefore must be included in the incorporation of privacy-protective solutions to assure the user that the information collected is secured [43].

The application of WBT in education requires training and support for educators to effectively use the data generated by these devices [17,142]. Also, the cost of WBT and the availability of technical support may limit their widespread deployment and scalability in educational settings [143]; however, wearables are typically less expensive than, for instance, neuroimaging equipment, and recent trends suggest they are becoming cheaper and more widely accessible [144].

Despite these challenges, there are several trends in wearable biosensing technology that have the potential to improve education. These biosensors can provide valuable information about students’ physiological responses during learning activities, allowing for adaptive and personalized educational interventions [36,114]. Additionally, the integration of physical sensors, machine learning, multifunctional AI, and VR with wearable biosensors is promising to improve the capabilities of these devices and solve some of the challenges [127].

The development of WBT capable of monitoring and analyzing emotional responses in real time has the potential to revolutionize the field of education [135]. By understanding students’ emotional states, educators can adjust their teaching strategies to optimize engagement and learning outcomes [121,145]. The use of wearable biosensors in collaborative learning environments can facilitate peer-to-peer collaboration and improve the quality of classroom engagement [146].

Finally, due to the COVID-19 pandemic, different alternatives to continue school programs had to emerge to ensure that students continue with their studies. This is where the new modality of virtual education entered the scene.

In recent years, a large number of learners around the world have enrolled in MOOCs offered by various online platforms. MOOCs stand out among the most popular e-learning methods. In 2017, there were more than 58 million learners, 800 universities, and 9400 MOOCs on MOOC platforms and the leading MOOC, Coursera, has thirty million learners and 2700 different courses [95]. This shows the relevance of virtual education in the last decade, and with the COVID-19 pandemic, this e-learning tendency reached its peak [141]. In the realm of virtual education, MOOCs provide significant flexibility for learning, but there is room for improvement in course structures. Students often face challenges related to their levels of consciousness while participating in online courses; physiological monitoring and WBT utilization can assist in recognizing students’ performance patterns, for instance, via high blood pressures in chronic stress conditions or confusion detection from acquired EEG signal data [147].

Virtual reality environments have found applications in educational contexts, suggesting that immersive technologies of this kind can effectively facilitate learning. In recent years, the integration of psychophysiological methods with VR technology has emerged as a tool for objectively evaluating their impact on learning. Among these methods, EEG has gained significant traction due to its association with cognitive processing data [35,148]. One noteworthy finding in this field is that virtual scenarios provide an opportunity to apply learned concepts and techniques instantaneously, emulating real conditions effectively [33]. However, when dealing with factual information and a high memory workload, the comparison of physical versus virtual environments should always be taken into account.

Additionally, WBT assists in making a reality out of personalized learning. Algorithms for stress [75,76,77,78,81] or mental fatigue evaluations [64], as well as biofeedback systems [43,74], can aid in creating timely and customizable resources and services that support students’ education—or even their lifestyle [73]—according to their individual needs.

As every trend shows, their implementation implies challenges that need to be solved in order to be executed successfully. This is where biometrics can help to improve the quality of virtual education to assure that students receive the knowledge they should. Some studies have proposed the use of sensors and software to study the biometric behavior of students to measure their attention level, the presence of stress, or their pulse rate to identify specific behaviors in students [68,123]. Wearables are intertwined with technology-enhanced learning, a concept that explores scalability and data aggregation, carrying implications across various domains. More significantly, they introduce innovative approaches, devices, and techniques to enhance education [143].

## 5. Conclusions

Wearables and biometric signals are linked today due to technological advances in both fields. New devices are constantly being researched, designed, and distributed with the capacity to obtain a wider variety of biometric data more efficiently and with greater precision. As time goes by, devices are progressively becoming more cost-effective [144]. As stated in previous studies, biometric data allow us to accurately determine the state and behavior of a person considering the subject’s profile and description [44]. This paves the way for further exploration into novel realms of research due to the growing field of wearables.

This review includes a total of 140 WBT studies that discuss their implementation in academic environments. In the studies analyzed, various focal points are discerned, such as the examination of emotional and academic stress of students in class or exams [70,75,76,78], the development of the student as a whole [82,83,84,85], academic achievement and improvement in students [58], and the impact of the use of different teaching resources and techniques [49], among others.

WBT is employed to examine teaching and learning effectiveness through data collection, analysis, biofeedback strategies, and qualitative surveys. This review presents EEG as the predominant NPM used in education studies. Some studies utilize EEG independently or in conjunction with other biometrics such as EDA, eye tracking, ECG, EMG, and blood pressure. The analysis and interpretation of these data in classrooms aim to explore mental states, assess physiological constructs, and evaluate teaching effectiveness from a cognitive perspective. Some of these studies focus on examining various facets of students, including stress, motivation, flow state, concentration, and cognition. They observe the impact of these factors on academic performance and psychological well-being, employing different algorithms for these assessments [20].

As stated previously, data captured via high-tech devices have shed light on students’ behavior and performance in academic environments. This information gives professors insights into students’ academic performance, learning outcomes, and achievements [5]. The recent technique of computing and analyzing brain synchrony between students and professors has been shown to have an impact on a student’s performance and achievement in their academic pathway, and this tool gives professors a broader understanding of their class engagement. Providing this feedback to professors allows them to further tailor and adapt their teaching according to the needs of the class [62].

Nowadays, some educational institutions are adopting and exploring the use of biometrics in education [15], in which some of its applications are to predict the performance of a student, to personalize the student experience, and to improve the efficiency of e-learning systems. Finally, it is crucial to keep in mind that the projects analyzed make use of sensitive biometric data collected by WBT. For this reason, and as mentioned in Section 4, it is important to prioritize and look after the privacy of the students by ensuring that the data are appropriately protected to keep this sensitive information safe [43].

The research in this field ought to gravitate towards some approaches to develop educational models tailored to the unique learning requirements of each student or to generate better predictive algorithms to accurately forecast academic performance and learning needs. Another recommendation for future studies is the impact of brain synchrony between students and educators on academic outcomes, which could lead to more effective teaching methods. By closely analyzing the data collected using this approach, it could be possible to provide constructive feedback to both students and educators, thereby enhancing teaching and learning processes.

When discussing biometrics and wearable technology applied in educational settings, several research approaches were detected. These include the development of educational models tailored to the unique learning requirements of each student and the improvement of predictive algorithms to accurately forecast academic performance and learning needs. Using these technologies can provide details of the teaching or learning quality in academic programs from a physiological perspective. This is of great importance in cases where the evaluation of students’ learning and/or skills is complicated. As WBTs provide a physiological-based assessment of mental and cognitive states, they are expected to be increasingly used in the future academic context to provide a more complete evaluation of educational objectives.

## Figures and Tables

**Figure 1 sensors-24-02437-f001:**
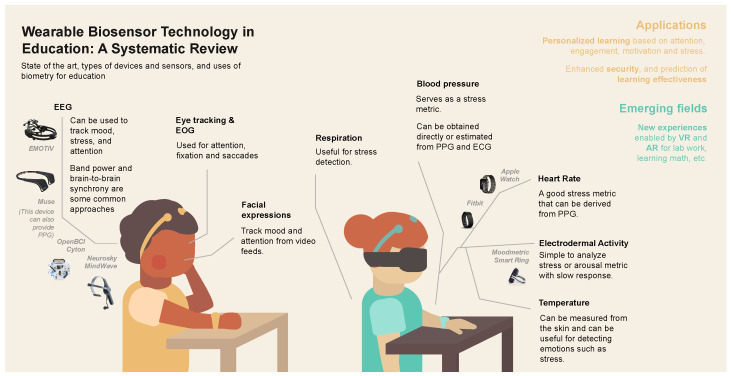
Graphical abstract for the present literature review. This figure provides a summary of the devices used to acquire each physiological measurement, and the use of each biometric in education is also explained.

**Figure 2 sensors-24-02437-f002:**
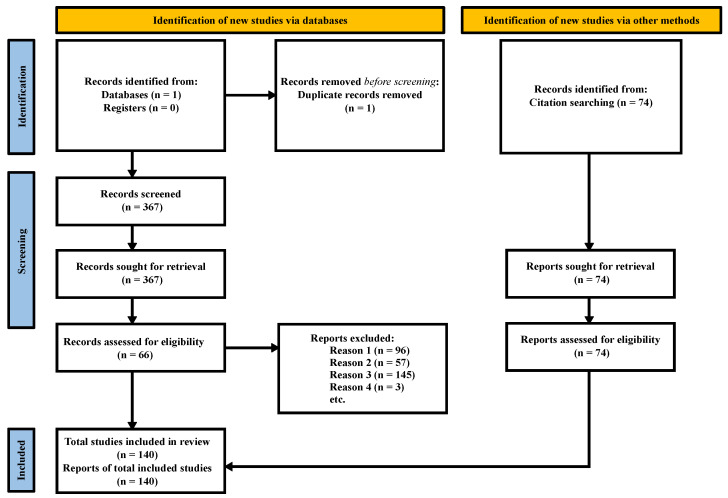
PRISMA flow diagram. The diagram shows the total works included in this review. The review was limited to one database (n = 1) and no registers (n = 0). In addition, 74 studies were identified through citation searching within Google Scholar.

**Figure 3 sensors-24-02437-f003:**
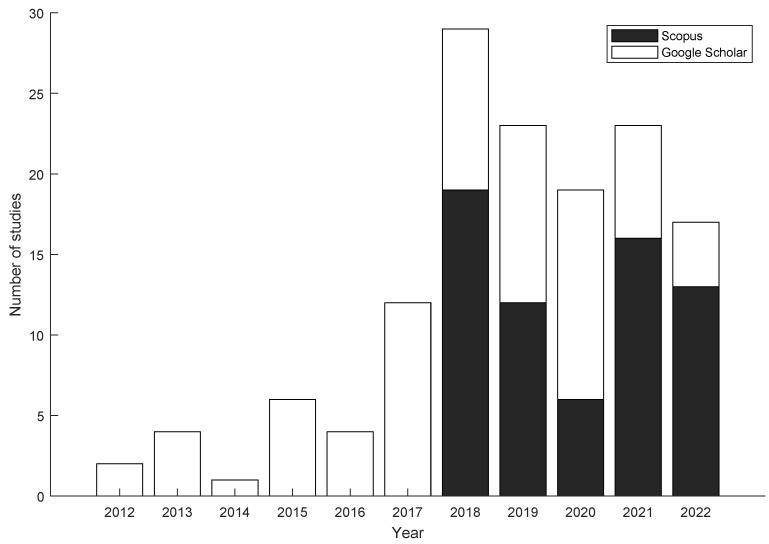
Temporal distribution of included studies from Scopus and Google Scholar.

**Figure 4 sensors-24-02437-f004:**
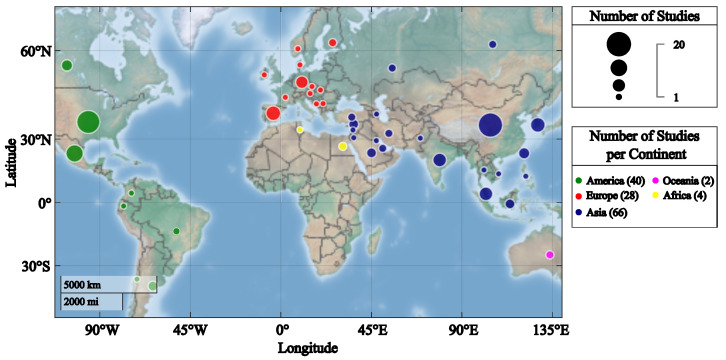
Geographical distribution of included studies from Scopus and Google Scholar.

**Figure 6 sensors-24-02437-f006:**
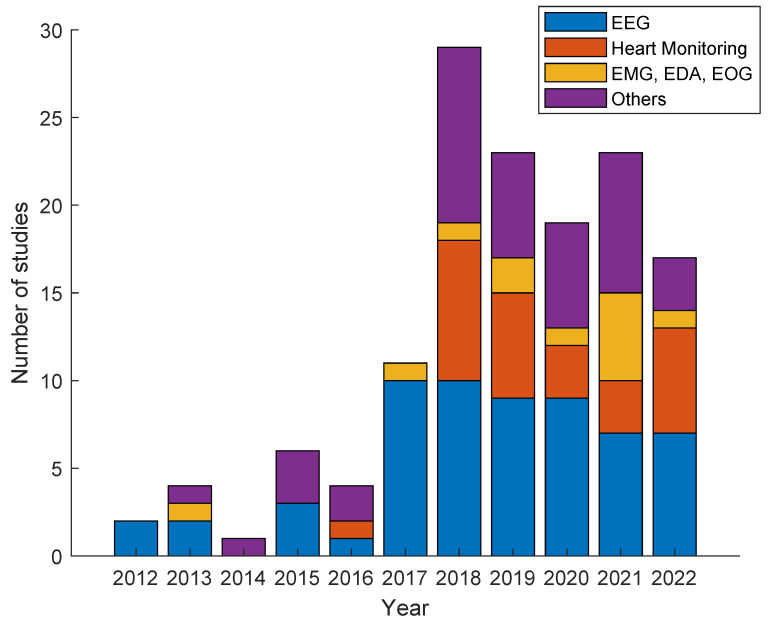
Temporal distribution of included studies divided according to their application.

**Figure 7 sensors-24-02437-f007:**
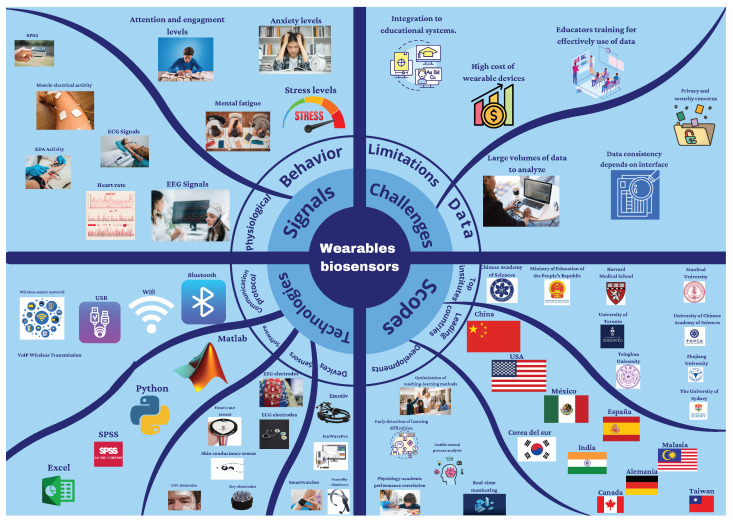
A graphical depiction of the results found in this review. It shows the countries, signals, devices, and institutions, among other characteristics, that are most present in the papers found.

**Table 1 sensors-24-02437-t001:** General characteristics of the included studies from Scopus.

Study	Objective	Education Type	Education Level	Institute	Country	Sample Size	Analysis Tools	Contribution
[42]	To determine stress levels in pharmacy students	Pharmacy education	University	Faculty of Pharmacy in Hradec Krávolé	Czech Republic	375 students	HRV, PSS-10, Statistics	Moderate stress levels while studying
[43]	To reduce children’s anxiety and stress	Academic education	Elementary school	Public school from the “Amara Berri” group	Spain	585 students	EmWave, BASC-S2, statistics	Biofeedback reduces students’ anxiety and stress
[44]	To evaluate sleep behaviors among college students	Academic education	University	Local university in South Korea	South Korea	86 students	Sleep behavior, Saliva sampling, HRV, GARS-K, statistics	Sleep behaviors are associated with stress
[45]	To investigate daily stress levels and EEG	Academic education	University	Suranaree University of Technology	Thailand	60 students	MSSQ, EEG, statistics	Stress among students alters brain functions
[46]	To analyze emotional stress in teachers	Academic education	University	Not provided	Japan	Not provided	EEG signals	Emotional stress recognition model for teachers
[47]	To develop a cost-effective monitoring device	STEM education	University	Not provided	China	Not provided	Arduino, Smartphone app, ECG signals	Cost-effective ECG signal testing device
[48]	To evaluate psychological stress in students	Academic education	University	Not provided	China	90 students	Classification algorithm, RBFNN and IELM	Importance of stress detection in education
[49]	To test technology in Korean teaching	Language education	University	Korean major in a university	China	50 students	Wireless sensing technology, tests	Impact of sensing technology in education
[50]	The use of wearables in the teaching and learning of English	Language education	University	Universiti Utara Malaysia	China	263 students	Statistics	Wearables can make learning easier by improving teaching themes, providing graphic teaching scenarios, and creating an overall independent teaching environment
[51]	To create scenarios for students to build confidence	Medical Education	University	Georgian College of Applied Arts and Technology	Canada	6 personal support worker students	Arduino, Bluetooth, vibration motor	Simulation enables students to reach learning outcomes
[52]	To integrate sensors and AR in EFL teaching	Language education	University	Zhejiang Yuexiu University	China	Simulation experiment	Sensors	AR is effective and can support English teaching
[53]	To investigate academic stress–achievement relationships	Medical education	University	Pusan National University School of Medicine	South Korea	97 students	HRV, statistics	Students with higher academic achievements have higher stress
[54]	To identify how sensors improve learning efficiency	Language education	University	Xingtai University, Universiti Teknologi Malaysia	China and Malaysia	Not provided	Machine learning, statistics	A classroom learning environment affected by the students’ movements allowed learning free from constraints
[55]	To detect students’ stress during the COVID-19 pandemic	Academic education	University	Engineering Department at the University of Pamplona	Colombia	25 students	Python 3.8, Tkinter library, ScikitLearn library	GSR resulted in the best NPM to identify stress
[56]	To propose a stress detection framework	Academic education	University	Not provided	Not provided	264 students and 32 police school students	Machine learning classification	Development of stress detection algorithms based on an adversarial transfer learning method and analysis of physiological signals
[57]	To use sensors in audio–visual language teaching	Language education	University	Speech and hearing research center of Peking University	China	4 subjects	MATLAB, classification	Line-of-sight change estimation classifier
[58]	To improve English language teaching by using sensors and VR	Language education	All education levels	Not provided	China	Not provided	Statistics	An online English teaching system via sensors/VR
[59]	To implement motor learning tools for students	Motor learning	Preschool	Not provided	Indonesia	65 students	Not provided	Measuring tool based on sensors to evaluate motor skills
[5]	To analyze teaching methods in basketball students	Physical education	University	Not provided	Not provided	108 students (49 women)	Statistics	Integration of micro classes and smart bands in a basketball course
[60]	To analyze stress in students during examination	Academic education	University	Sastra University	India	14 students	Statistics	Identification of higher stress before testing
[61]	To create a student authentication system for online learning	Online academic education	University	Moodle, Blackboard and OpenEdx	Latin America, Europe and Asia	350 students	Electron JS	An automated, online student authentication system
[26]	To create a real-time detection system of students’ flow state through EEG	Academic education	Elementary school	Department of Science Education, National Taipei University of Education	Taiwan	30 students	BCS, MOM-tp, Statistics	Future e-learning development with a BCI system
[62]	To motivate students with AI to improve their perfomance	Academic education	University	Not provided	Not provided	4 students	Statistic, HRV, Grovi Pi Sensors, Raspberry Pi	Introduction of the Education 4.0 Framework
[20]	To find links between physiological measurements, obtained with IoT devices, and students’ concentration	Academic education	University	University of Novi Sad	Serbia	15 students	Apple Watch, eye tracker, canvas, statistics	A higher HR correlates to lower concentration levels.
[63]	To find cognitive-wise growth in mobile device use in the classroom	Academic education	University	National Institute of Technology Agartala	India	58 students	EEG Headset, survey, statistics	Use of mobile devices in classrooms to enhance the quality of education
[64]	To analyze mental fatigue conditions in the occipital region	Academic education	High school	Senior High School 2 Malang	Indonesia	13 students	EEG Headset, questionnaire, statistics	Mental fatigue is a life-threatening factor in high-school students
[65]	To study changes in stress patterns during tests	Academic education	University	Ganja State University	Azerbaijan	68 students	EEG, Excel, SPSS	Reference physiological values are needed for studying stress patterns in education
[66]	To demonstrate the influence of AR in concentration	Technological education	University	Federal University of Rio Grande do Sul	Brasil	5 students	AR, EEG headset, platforms	Increased student attention during AR interaction
[67]	To solve missing data problems and human stress level prediction	Academic education	University	Not provided	Not provided	75 students	Smart-wristband data, MATLAB	Method for solving missing data problems through data completion with diurnal regularizers and temporally hierarchical attention network methods
[68]	To recognize students’ exam stress levels	Academic education	University	University of Tuzla	Bosnia and Herzegovina	10 students	BITalino, MATLAB, machine learning	Wearables can be used for building automated stress detection systems
[69]	To test the effects of time limitation on exam performance	Academic education	University	Institute of Space Technology, Islamabad	Pakistan	14 students	EEG signals	Performance deteriorates during timed tests
[70]	To measure academic stress to provide better ways to cope with it	Academic education	University	University of Turku	Finland	17 students	Smart device measures stress via physiological signals	Relation between study-related and non-study-related stress
[71]	To use EEG to measure e-learning effectiveness	Academic education	Kindergarten	Tadika Advent Goshen Kota Marudu, Pacos Trust Penampang, Pusat Minda Lestari UMS Kota Kinabalu	Malaysia	98 students and 6 teachers	Effective learner application for EEG, and a mobile learning app	E-learning success is best judged in short sessions with suburban children
[72]	To measure HRV changes in students during different stages of an exam	Academic education	University	Lebanese University	Lebanon	90 students	HR, SDNN, RMSSD, pNN50, LF, HF, LF/HF	Gender differences during assessment of stress in real exams
[73]	To find statistical differences between lifestyles and stress levels	Academic education	University	American University of Madaba	Jordan	19 students	GRS data, Microsoft Band 2, mobile app, online survey	Correlations were found between GSR values and physical activity level
[74]	To review the learning behavior with biofeedback	Academic education	University	Not provided	China	106 students	EEG headset, eye tracker, statistics	Improving learning efficiency in autonomous learning settings is essential
[75]	To evaluate the psychological state of college students under test stress	Academic education	Junior college	Not provided	Not provided	15 students	MATLAB, EEG, neural networks, test questions	Students with higher test stress are more likely to face psychological health problems
[76]	To compare students stress appearing for previva/postviva during exams	Medical education	University	Navodaya Dental College and Hospital	India	70 students	Statistics, mobile app, Smartphone	Academic examinations produce situational stress in students and result in anxiety
[77]	To study stress-reduction techniques during microteaching in preservice teachers	Academic education	University	Not provided	Not provided	100 teachers	HR, blood pressure, statistics	Biofeedback was not effective to reduce stress in this sample of preservice teachers
[78]	To evaluate solutions for stress in students using COTS wristbands	Academic education	University	University of Vigo	Spain	12 students	COTS wristbands, machine learning, lectures	A protocol to evaluate student stress in classrooms based on HR, temperature, and GSR
[79]	To understand interactions with a visual search interface	Academic education	All education levels	Not provided	Not provided	20 students	EEG signals, E-prime 2, EEGO, ASA, Minitab17, ANOVA, Statistics	EEG experiment can be used as a basis to judge cognitive errors
[80]	To study how wearables support learning activities and ethical responsibilities	Academic education	All education levels	Oslo Metropolitan University	Norway	Not provided	Wearables	Wearables in teaching and learning provide pedagogical opportunities
[81]	To monitor stress levels during exams in students	Academic education	University	Universidad del Magdalena, Universidad del Norte	Colombia	20 students	EEG Emotiv Insight	A desktop app that monitors stress according to parameters obtained from EEG signals and the Emotiv Insight Software
[82]	To help teachers with wearables to collect data and provide feedback	Academic education	Elementary school	An elementary school in Zhaoqing City	China	Not provided	Wearable device	A model to collect data and give feedback
[83]	To help students with intellectual disabilities to learn	Academic education	All education levels	Middle East Technical University	Turkey	4 students	Wearable clothing	A way to help people with disabilities by creating an app and plushies with smart clothing that facilitate the learning of internal body organs
[84]	To improve the quality of teaching micro technology	Academic education	University	Technische Universität Ilmenau	Germany	30 students	Smart watch, fitness tracker, EEG, EMG	Techniques in the design process through formative evaluation
[85]	To analyze human motivation and efficacy processes	Academic education	University	St. Petersburg State University‘s Psychology Faculty	Russian	20 students	Biofizpribor, ECG	Improved educational and therapeutic interventions

**Table 2 sensors-24-02437-t002:** General technical characteristics of the included studies from Scopus.

Study	Sensor	Biometry Device	Sim or Exp	Communication Protocol	Type of Storage	Computing Engine	Processing	Software	Qualitative Index	Quantitative Index	Study Outcome
[42]	Infrared PPG ear sensor	EmWavePro (HeartMath Inc., Boulder Creek, CA, USA)	Experimental	Not provided	Not provided	No	Statistics	Kubios HRV (Kubios, Kuopio, Finland)	PSS-10, sociodemographic data	Total power, VLF, LF, HF, LF/HF, SDNN, Coherence5	No significant changes in PSS-1O and HRV
[43]	Non-invasive auditory sensor	Not provided	Experimental	USB	No	No	Statistics	EmWave 2021 Pro. Version (HeartMath Inc, Boulder Creek, CA, USA)	BASC II test	HRV	Students learned to breathe consciously
[44]	Heart rhythm scanner PE	Octagonal motion logger Sleep Watch-L (Ambulatory Monitoring, Ardsley, NY, USA)	Experimental	Not provided	Not provided	No	Statistics	Action W-2, IBM SPSS Statistics version 25 (IBM, Armonk, NY, USA)	GARS-K	Saliva, HR, SD, SDNN, LF/HF	sAA and HRV are significant in sleep disorders
[45]	EEG electrodes	Not provided	Experimental	Not provided	Not provided	No	Statistics	IBM SPSS Statistics version 17	MSSQ, sociodemographic data	EEG signals	Stress analysis improves classes
[46]	EEG electrodes	Not provided	Experimental	Not provided	Not provided	No	DFA, Linear Feature Selection, statistics	Not provided	Not provided	EEG signals	Deep learning for emotion recognition
[47]	AD8232 ECG chip	Not provided	Experimental	Bluetooth HC-05	Not provided	No	Signal filtering	Not provided	Not provided	HRV	System that facilitates HRV analysis
[48]	EEG electrodes	Not provided	Experimental	Not provided	Not provided	No	AdaBoost, RBFNN, IELM	Not provided	Sociodemographic data, self-evaluation	EEG signals	Algorithm with excellent accuracy
[49]	EEG electrodes	Not provided	Experimental	Wireless communication	Internet and satellite	No	Statistics	Not provided	Not provided	Not provided	Wireless sensors can improve student grades
[50]	Not provided	Not provided	Experimental	Not provided	Not provided	No	Statistics	IBM SPSS Statistics version 13.0	Not provided	Not provided	Wearable use is associated with better test scores
[51]	Arduino MKR1010, vibration motor	Not provided	Experimental	Bluetooth and visual via website	Not provided	No	Statistics	Arduino (Arduino, Lombardia, Italy)	Not provided	Not provided	Wearables provided insight into a medical scenario
[52]	Track movement, heartbeat, trajectory	Not provided	Simulation	High-bandwidth optical fiber technology	Not provided	No	Survey summary and statistics	Not provided	Not provided	Temp, Disp, RS, MF, Stress, Vibration	AR supports the practice of English teaching
[53]	Not provided	SA2000E HRV analytic equipment	Experimental	Not provided	Not provided	Not provided	Statistics	IBM SPSS Statistics 24.0	Socio-demographic data	BMI, HRV, SDNN, LF, HF, LF/HF	Women suffer more academic stress than men
[54]	Light and temperature sensors	Not provided	Experimental	WiFi	Not provided	Not provided	Machine learning	Not provided	Satisfaction survey	Light and temperature	Students approve of the system
[55]	GSR sensor, MOX gas sensors, electrodes	GSR, ECG, EMG, Electronic Nose System	Experimental	I2C, Wifi	Not provided	No	LDA, KNN, SVM	Python 3.8, (Python Software Foundation, Beaverton, OR, USA), Raspbian environment	SISCO Inventory	HRV of ECG, GSR, gas sensors’ response, EMG	GSR data were best in relaxed and stressed states
[56]	EDA, PPG, ST, ACC sensors	Wrist-worn wearable device	Experimental	Bluetooth	Not provided	No	SVM, KNN	Python	Self-reported stress levels	Mean, SD, HRV, BPM, IBI, LF, HF, Average	Classification of stressed and relaxed states
[57]	Heog, NEMG, and IMU sensors	NeuroScan synamps 2 system	Experimental	Not provided	Not provided	No	Window slicing, FCN, LSTM and SVM	MATLAB (The Mathworks Inc., Natick, MA, USA)	No	Heog Value, NEMG amplitude and RMS	Estimation of change angle of line of sight
[58]	Odometer, Polaroid 6500 sonar modules	Milodometer and Sonar systems	No	Not provided	Not provided	No	SIFA, KF, statistics	Not provided	No	Skeleton position, movement, rotation angle	VR for an online English teaching experience
[59]	Movement sensor	Limit switch sensor	Experimental	Not provided	Not provided	No	No	Not provided	Scoring of motor ability	Time between movements	A motor skills test tool from the locomotor component
[5]	Heart rate and blood pressure sensors	Smart Redmi bracelet (Xiaomi, Beijing, China)	Experimental	Wireless sensor network	Not provided	Semantic mobile computing	Statistics	IBM SPSS Statistics 17.0	No	Scores of physical exercises, P value	Better student performance in basketball classes
[60]	Dry EEG electrodes	Enobio system (Neuroelectrics, Barcelona, Spain)	Experimental	Not provided	Stored in the computer	Not provided	WPT, Statistics	PSYTASK (Bio-Medical Instruments Inc., Detroit, MI, USA), ENOBIO NIC 1.4 (Neuroelectrics, Barcelona, Spain)	Arithmetic task	EEG relevant alpha and theta component energy	Students were highly stressed before examination
[61]	Microphone, webcam, keyboard	Proctoring system	Experimental	VoIP	DB	Cloud	FaceBoxes, M3L, NNs, Kaldi	Electron JS (OpenJS Foundation, San Fransisco, CA, USA)	User experience test	Images, audio, keystroke dynamics	Better biometric models are needed
[26]	Mobile dry EEG sensors	NeuroSky MindWave Headset (NeuroSky, San Jose, CA, USA)	Experimental	Not provided	Not provided	No	Average, EEG power, statistics	SPSS 20.0 (SPSS Inc., Chicago, IL, USA), Microsoft Excel (Microsoft, Redmond, WA, USA), WEKA 3.8 (University of Waikato, Hamilton, New Zealand)	SR-F	EEG signals	EEG-F detects flow experience
[62]	PPG, Grove Pi sensors	Smartphone, Raspberry Pi (Raspberry Pi, Cambridge, UK), Smartwatch	Experimental	I2C, Wifi, Bluetooth	Not provided	Google Cloud TTS	Statistics	Python, ECG for Everybody	Sound	HRV, Temp, Cal, Hum, Steps	Relation between self-test and biosignals
[20]	HR and eye tracking sensor	Apple Watch (Apple, Cupertino, CA, USA)	Experimental	Not provided	Health Mobile App	Cloud	Statistics	Not provided	Quiz evaluation	Heart rate	Initial HR in the quiz affects concentration
[63]	Mobile dry EEG sensors	NeuroSky MindWave Headset	Experimental	Not provided	Not provided	No	ThinkGear ASIC, statistics	JASP 0.10.2 (JASP Statistics, Amsterdam, The Netherlands)	Survey	EEG signals	Bayes factor supports that mobile devices have positive effects in class
[64]	EEG electrodes	EMOTIV EPOC+ (EMOTIV, San Francisco, CA, USA)	Experimental	Bluetooth	Not provided	No	MAV and SD	Not provided	IFS	EEG signals	8 h school days can cause mental fatigue
[65]	EEG electrodes	Not provided	Experimental	Not provided	Not provided	No	Statistics	IBM SPSS Statistics, Microsoft Excel	Not provided	EEG signals	Differences in brain signals between 1st and 5th year students
[66]	Mobile dry EEG sensors	NeuroSky MindWave Headset	Both	Bluetooth	Student’s inventory	No	Statistics	Moodle (Moodle, West Perth, WA, USA), Unity 3D (Unity Technologies, San Francisco, CA, USA), Vuforia (PTC, Boston, MA, USA)	Self-reported attention levels	EEG signals, attention levels	High concentration with AR app
[67]	Sleeping, walking, running, and cycling sensor data	Smart-wristband	Experimental	Not provided	Not provided	No	Machine learning	MATLAB, Tensorflow (USENIX Association, Berkeley, CA, USA)	Online survey	Data from smart-wristband	Data filling and stress level prediction
[68]	EDA and ECG sensors	BITalino	Experimental	Bluetooth	Not provided	No	Statistics, KNN, SVM, LDA	MATLAB	Not provided	ECG and EDA signals	SVM was the most accurate with 91%
[69]	EEG electrodes	OpenBCI Cyton (OpenBCI, Brooklyn, NY, USA)	Experimental	Wireless transmission	At the device level	No	Mean and SD of PSD	MATLAB and EEGLAB (Swartz Center for Computational Neuroscience, San Diego, CA, USA)	Math test	EEG signals	Stress increases in timed exams
[70]	EDA sensor	Moodmetric smart ring	Experimental	Not provided	Not provided	No	Statistics	Microsoft Excel	Written diary	EDA signal	Correlation between non-study and studying
[71]	EEG electrodes	NeuroSky MindWave Headset	Experimental	Not provided	Not provided	No	Statistics	Mobile learning application	Questionnaire	EEG signals	Suburban students tend to learn more with m-learning
[72]	Ambu WhiteSensor WS electrodes	Cardio Diagnostics (Cardio Diagnostics Inc., Chicago, IL, USA)	Experimental	Not provided	Not provided	No	Statistics	Kubios HRV 2.2	Questionnaire	HRV parameters	HRV in females is lower before/after examination
[73]	GSR sensor	Microsoft Band 2 (Microsoft, Redmond, WA, USA)	Experimental	Bluetooth	Mobile app	No	Statistics	Not provided	Online survey	GSR data	GSR data are dependent on human behavior
[74]	Mobile dry EEG sensors, eye tracker	NeuroSky MindWave Headset	Experimental	Not provided	Not provided	No	Statistics	Minxp, IMB SPSS Statistics 19	Bloom’s taxonomy survey	EEG signal	Biofeedback may act as a metacognitive method
[75]	EEG electrodes	Not provided	Experimental	Not provided	Not provided	No	Neural networks	MATLAB	Test questions	EEG signals	EEG signals are multi-fractal signals
[76]	HR, oxygen and stress sensors	Smartphone Samsung S7 (Samsung, San Jose, CA, USA)	Experimental	Not provided	Mobile app	No	Statistics	Android S-HEALTH (Android, Palo Alto, CA, USA)	Not provided	HR, oxygen saturation, stress levels	Gender differences in stress aptitude
[77]	HR, blood pressure sensors	HeartMath EmWave, GE Dinamap PRO 400 Vitals	Experimental	Not provided	Not provided	Not provided	No	Statistics	Online survey	HR and blood pressure data	No differences in stress levels after microteaching
[78]	HR, ST, GSR, ACC sensors	Wristband	Experimental	Bluetooth	Server’s database	No	Machine learning	Not provided	Quiz and lecture sessions	Information from wearable	Average classification accuracy of 97.62%
[79]	EEG electrodes	Not provided	Experimental	Not provided	Not provided	No	ANOVA, statistics	E-prime 2 (Psychology Software Tools, Pittsburgh, PA, USA), EEGO (ANT Neuro, Hengelo, The Netherlands), ASA, Minitab17 (Minitab LLC, State College, PA, USA)	Not provided	EEG signals	N200 is produced by visual attention
[80]	GPS and HR	Fitbit Surge (Fitbit, San Francisco, CA, USA)	Experimental	WiFi	Computer storage	Cloud	Statistics	Microsoft Excel	Not provided	Location and pulse data	Wearables are not yet ready for use in teaching and learning
[81]	EEG electrodes	EMOTIV Insight (EMOTIV, San Francisco, CA, USA)	Experimental	Bluetooth Smart 4.0	Not provided	No	Not provided	Microsoft Excel, SDK of EMOTIV Insight (EMOTIV, San Francisco, CA, USA)	Test IDARE	EEG signals	Increased stress in both subjects
[82]	HR sensor	Love buckle health (CoCoQCB2)	Experimental	Bluetooth	System platform	Server	Statistics	Not provided	RPE scale	Heart rate	Measured data should be more accurate
[83]	Not provided	Not provided	Experimental	Not provided	Not provided	No	Not provided	App	Position of organs	Not provided	Students learned organ locations
[84]	EEG, ECG, EDA, EMG, HR, BP, BG, BO sensors	Not provided	Experimental	Not provided	Not provided	No	Not provided	Not provided	SR-F	EEG, ECG, EDA, EMG, HR, BP, BG, BO	E-learning system prototype
[85]	EEG and ECG electrodes	Not provided	Experimental	Not provided	Not provided	No	Statistics	Not provided	FAM test	EEG and ECG signals	Stress was related to poor answers

**Table 3 sensors-24-02437-t003:** Biometry devices used in the included studies from Scopus.

Biometry Device	Signal	Sensing Device	Communication Protocol	Type of Data Storage	Power	Studies
EmWavePro	HRV	PPG, ear sensor	USB	Software	Rechargeable lithium-ion battery	[42,77]
Octagonal motion logger sleep Watch-L	Not provided	Not provided	Serial communications (COM) port	2 Mb of non-volatile memory	Power supply, changeable batteries	[44]
SA2000E HRV analytic equipment	HRV	Not provided	Not provided	Not provided	Not provided	[53]
NeuroScan synamps 2 system	EEG	EEG Electrodes	USB 2.0	Neuroscan software	120V AC	[57]
Smart Redmi bracelet	Heart rate, blood pressure, oxygen saturation	6-axis sensor: 3-axis accelerometer and 3-axis gyroscope, PPG heart rate sensor and light sensor	Bluetooth low energy	App	200 mAh	[5]
Enobio system	EEG	Wet, semi-dry and dry electrodes	WiFi or USB	MicroSD or Software	Rechargeable system using Li-Ion battery	[60]
NeuroSky MindWave Headset	EEG and ECG signals	12 bit raw brainwaves and power spectrum, eSense, sensor arm up and down	BT/BLE dual mode module	App	AAA battery	[26,63,66,71,74]
Raspberry Pi	Not provided	GPIO to connect sensors	SSH, UART, I2C, SPI, USB, LAN, WIFI, Bluetooth	DAS, NAS	1.8 a 5.4 W	[62]
Apple Watch	Heart rate, blood pressure, oxygen saturation, movement	PPG heart rate sensor, light sensor, 3-axis accelerometer, 3-axis gyroscope	Bluetooth	DAS, NAS, App	Rechargeable lithium battery	[20]
EMOTIV EPOC+	EEG signals	9 axis sensor: 3-axis accelerometer, 3-axis magnetometer. EGG sensors.	Bluetooth low energy	Software	Internal 640mAh lithium-polymer battery (rechargeable)	[64]
BITalino	ECG, EMG, EDA, and EEG signals	MCU, Bluetooth, Power, EMG, EDA, ECG, Accelerometer, LED, and Light Sensor	Bluetooth 2.0 + EDR or Bluetooth 4.1 BLE, Bluetooth (BT) or Bluetooth low energy (BLE)/BT dual mode	OpenSignals Software	Battery: 700 mA 3.7 V LiPo (rechargeable)	[68]
OpenBCI Cyton	EEG, EMG, ECG	Not applicable—it serves as a connection between sensors	BLE, USB dongle via RFDuino radio module	PC, mobile device	3–6 V DC	[69]
Moodmetric smart ring	EDA	Not provided	Bluetooth Smart	Moodmetric app and Moodmetric cloud	Internal, non-removable, rechargeable Li-Ion battery	[70]
Cardio Diagnostics	ECG	Transmitter adhesive patch	Not provided	Cloud	Rechargeable battery	[72]
Microsoft Band 2	ECG and temperature	Optical sensor, three-axis accelerometer, gyrometer, galvanic skin sensors and skin temperature sensor.	Bluetooth 4.0	Not provided	Charge by a 200 mAh Li-polymer battery.	[73]
Smartphone Samsung S7	Heart rate and oxygen saturation	SpO2 and heart rate sensor	Not provided	Samsung S-health software	Rechargeable Li-Ion battery	[76]
GE Dinamap PRO 400 Vitals	Blood pressure, temperature, oxygen saturation	Blood pressure cuff, SpO2 sensor, oral temp sensor	Remote operation with DINAMAP® Host Communications Protocol	Not provided	DC input, battery power, host port power	[77]
Fitbit Surge	ECG	An MEMS 3-axis accelerometer and optical heart rate tracker	Bluetooth 4.0	fitbit.com dashboard	Rechargeable lithium-polymer battery.	[80]
EMOTIV Insight	EEG signals	EEG semi-dry sensors, IMU, accelerometer, gyroscope, magnetometer	Bluetooth low energy	Not provided	480 mAh battery	[81]
Love buckle health (CoCoQCB2)	Heart rate	Not provided	433 MHz radio, Bluetooth	App, server	Not provided	[82]

## Data Availability

Not applicable.

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
