# Peer review of "Wearable Biosensor Technology in Education: A Systematic Review"

_sensors, 2024, doi:10.3390/s24082437_

Round 1

Reviewer 1 Report

Comments and Suggestions for Authors

The manuscript aims to provide a comprehensive review of the utilization of Wearable Biosensor Technology (WBT) in educational settings over the past decade, while also offering insights into its evolution, integration into addressing educational challenges, and potential future research directions. The authors have meticulously followed the PRISMA guidelines, providing detailed information regarding the search strategy, inclusion/exclusion criteria, and evaluation of relevant literature, thus enhancing the transparency and rigor of this review.

However, despite the manuscript's well-structured and thoroughly researched nature, it appears to serve more as a review piece and an incomplete textbook tutorial, lacking in offering significant new insights. Additionally, the article's relevance to sensor technology seems limited. For example, discussing different types of sensors involved in WBT and conducting a simple analysis of the principles of them could be a suitable approach.

Overall, with some enhancements in offering novel insights and aligning with the designated theme, the manuscript has the potential to significantly improve its contribution to the literature on WBT in educational settings.

Comments on the Quality of English Language

The manuscript effectively elucidates the research methodology, the evolution of WBT technology, and its applications in the field of education, with language that is clear and easy to read.

Reviewer 2 Report

Comments and Suggestions for Authors

I have the following suggestions:

1) There are several abbreviations used in the paper. I suggest making a section related to the abbreviations and their full form at the end of the paper. This will help the reader to go through the paper easily. 

2) From table 1 and table 2, why the Scopus database has been selected? Why not the Web of Science database which is more authentic or widely used than these databases? 

3) there should be no new references or citations in the conclusion section. This section is dedicated to concluding and summarizing the study provided in previous sections. Therefore, I suggest the authors remove those references from the conclusion section. 

4) Are the devices presented in this paper continuous monitoring devices (24/7) or they are worn for a specific period and gather the information? Please clarify this in the paper. 

5) In the paper, the emphasis is given to the number of publications, who published it, the country of origin, Scopus etc. And less attention is given to the real applications or commercial availability of such devices in the education sector. I would suggest the authors focus on the real-time applications of these wearables instead of just focusing on the publications etc. 

6) The author should also comment on the accuracy of the measurements taken by the proposed wearables as suggested in table 1. What is the general outcome of those devices proposed or experiments? 

Comments on the Quality of English Language

none

Round 2

Reviewer 2 Report

Comments and Suggestions for Authors

I am willing to accept the paper in its current form. 

Comments on the Quality of English Language

none.